# Common Polymorphisms in the Glycoproteins of Human Cytomegalovirus and Associated Strain-Specific Immunity

**DOI:** 10.3390/v13061106

**Published:** 2021-06-09

**Authors:** Hsuan-Yuan Wang, Sarah M. Valencia, Susanne P. Pfeifer, Jeffrey D. Jensen, Timothy F. Kowalik, Sallie R. Permar

**Affiliations:** 1Department of Pediatrics, Weill Cornell Medicine, New York, NY 10065, USA; hsuanyuan.wang@duke.edu; 2Duke Human Vaccine Institute, Duke University Medical Center, Durham, NC 27710, USA; sarah.valencia@duke.edu; 3Center for Evolution & Medicine, School of Life Sciences, Arizona State University, Tempe, AZ 85281, USA; susanne.pfeifer@asu.edu (S.P.P.); jeffrey.d.jensen@asu.edu (J.D.J.); 4Department of Microbiology and Physiological Systems, University of Massachusetts Medical School, Worcester, MA 01655, USA; timothy.kowalik@umassmed.edu

**Keywords:** human cytomegalovirus, glycoprotein, polymorphism, genotype, strain-specific immunity

## Abstract

Human cytomegalovirus (HCMV), one of the most prevalent viruses across the globe, is a common cause of morbidity and mortality for immunocompromised individuals. Recent clinical observations have demonstrated that mixed strain infections are common and may lead to more severe disease progression. This clinical observation illustrates the complexity of the HCMV genome and emphasizes the importance of taking a population-level view of genotypic evolution. Here we review frequently sampled polymorphisms in the glycoproteins of HCMV, comparing the variable regions, and summarizing their corresponding geographic distributions observed to date. The related strain-specific immunity, including neutralization activity and antigen-specific cellular immunity, is also discussed. Given that these glycoproteins are common targets for vaccine design and anti-viral therapies, this observed genetic variation represents an important resource for future efforts to combat HCMV infections.

## 1. Introduction

Human cytomegalovirus (HCMV) is highly prevalent, with an estimated 83% seroprevalence in the global population [1]. While HCMV-infected immunocompetent individuals are generally asymptomatic, HCMV can cause morbidity and mortality for immunocompromised individuals [2], including in the setting of organ transplant recipients, acquired immunodeficiency syndrome (AIDS), and congenital infection. Infections with multiple HCMV strains are commonly observed among these groups [3,4,5], and strain replacement has additionally been noted [6]. Furthermore, frequent recombination between strains continuously generates novel genotypic combinations [3]. These clinical observations indicate the interesting complexity of the population genetic environment in HCMV [4], and hence elucidating the evolutionary mechanisms dictating observed variation remains as a major research topic in the pursuit of effective vaccines and anti-viral therapies.

## 2. Complexity of HCMV Genomes and Viral Populations In Vivo

HCMV has long been recognized as being genetically variable. In 1980, Huang et al. observed differences in restriction fragment length polymorphisms (RFLPs) in HCMV samples cultured from congenitally infected infants [5]. Likewise, different laboratory strains were recognized, some of which were utilized in early vaccine trials [7,8]. As DNA sequencing technologies developed, entire genomes of HCMV could be delineated, highlighting nucleotide-level differences between strains on the order of ~2% [9,10,11,12]. Subsequent short read, deep-sequencing studies confirmed genome-wide differences in HCMV consensus sequences and revealed the wide-spread presence of rare alleles, insertions, and deletions in patient samples [13,14,15,16]. Indeed, the significance of this within-host population-level variation in HCMV was quickly recognized [14,17,18,19], with nucleotide diversity approaching that of certain RNA viruses [20]. As with RNA viruses, this diversity allows for populations to be structured and varied, driven by both stochastic (e.g., infection bottleneck) and more deterministic (e.g., natural selection) evolutionary forces [15,21]. The combination of constant purifying selection and episodic positive selection, together with population size change and the related effects of genetic drift, has been found to result in a relatively high-level of neutral and weakly deleterious variation in patient samples [10]. Unlike RNA viruses however, mutation rates tend toward the more modest range of ~2 × 10^−7^ per site per replication [17]. Other factors contributing to the observed population-level diversity include compartmentalization and gene flow, post-infection population growth rates, reinfection, and recombination [17,19,22,23,24]. Collectively, these features result in a collection of genetic variants in clinical specimens that may influence disease severity, responses to antiviral therapies, and vaccine candidate design.

## 3. Common Polymorphisms in the Glycoproteins of HCMV

The ~235 kb double-stranded DNA genome of HCMV encodes approximately 230–250 putatively functional open reading frames (ORFs) [12,17,25]. These ORFs include up to 65 unique glycoproteins [9,26], and several of them are responsible for viral attachment to and entry into the host cells. Due to their pivotal role in initiating signaling transduction cascades in target cells and propagating HCMV infection, glycoproteins have been identified as key HCMV vaccine targets. Interestingly, instead of functioning alone, glycoproteins on HCMV virions form complexes, which facilitate the pathogenesis of HCMV infection. These complexes include the glycoprotein B oligomer (gB) [27], glycoprotein M/glycoprotein N (gM/gN) dimer [28], glycoprotein H/glycoprotein L/glycoprotein O (gH/gL/gO) trimer [29], and gH/gL/gO/UL128-130 pentameric complex [30]. Table 1 summarizes the commonly observed polymorphisms in these glycoproteins, including the length and genotypes, as well as the most variable regions described to date. In order to determine the most variable regions among each glycoprotein, multiple sequence alignments were generated using T-coffee version 13.45.0.4846264 [31] (Appendix A). In addition, Table 2 presents the genotypes of several common laboratory-adapted HCMV strains, including the strains Towne, TB40E, AD169, Toledo, VR1814, and Merlin.

These highly variable sites in the glycoproteins, together with viral cytokine/chemokine proteins (human cellular homologs), are commonly considered to be key factors for host immunity and HCMV infection [45]. This suggests strain-specific immunity, defined as the discrete immune responses elicited by different variants of the virus genomes. Strain-specific immunity is usually classified by the strain-specific humoral and antigen-specific T-cell responses. With regards to humoral immunity, strain-specific neutralization of HCMV has been identified in human sera [46], and monoclonal antibodies isolated from both humans and rabbits have also demonstrated strain-specific recognition [47]. Strain-specific T-helper-cell response to gH has also been reported when comparing the T-cell proliferative response to antigens from strains Towne and AD169 [48]. Given that strain-specific immunity is associated with observed polymorphisms, we review these commonly variable glycoprotein sites, summarize their current geographic distributions, and present evidence relating this diversity with immune response.

## 4. gB

HCMV gB is the best-characterized glycoprotein to date and is encoded by UL55. Consisting of ~900 amino acids (NCBI accession number: YP_081514), gB is composed of a large ectodomain region, membrane-proximal region, transmembrane domain, and cytoplasmic domain [49]. The prototypic gB undergoes proteolytic cleavage approximately at codon 460 which generates two gB polypeptides, gp116 and gp55 (also known as gp58), that are covalently linked by disulfide bonds. While most of the gB ectodomain belongs to gp116, gp55 is a type1 transmembrane protein settled on the HCMV envelope. Required for HCMV entry into target cells and HCMV infection via cell-to-cell spread, gB is highly glycosylated and genetically variable [27]. The glycosylation sites of gB include 18 possible N-linked glycosylation sites and two O-linked glycosylation sites [50,51]. In addition, the polymorphisms within gB genotypes occur mostly around the gp55 cleavage site at codon 460 as well as, to a lesser extent, around several coding regions of gp116—an observation which has been associated with homologous recombination [32,33,52,53]. Due to the intragenic variations within the gB sequence, gB genotypes have been defined based on the variations within C-terminus, N-terminus [54,55,56], and the gp55 cleavage site [57] as originally proposed by Chou et al. [32]. Interestingly, samples collected at multiple body sites from the same patient can display different gB genotypes, suggesting distinct cell tropism of HCMV, consistent with the compartmentalization of viral populations observed by deep sequencing [15,17].

To date, five different gB genotypes (gB-1, gB-2, gB-3, gB-4, and gB-5 previously referred to as gB-3′ [34,58,59]) have been identified [32,33]. We here review previous research investigating the correlations between gB genotypes and clinical outcomes, focusing (by and large) on individuals with compromised immune systems (e.g., organ/bone marrow transplant recipients, AIDS patients, and congenitally or perinatally HCMV-infected infants). Figure 1 organizes the gB genotyping studies alphabetically by study continent and country, as well as by patient profile. All five genotypes have been detected in Asia, Europe, and North America, however their geographic distribution differs: gB-1 is the most prevalent genotype in Asia and Egypt [60,61,62,63,64,65,66,67,68,69,70,71,72,73,74,75,76,77,78,79,80,81,82,83,84,85,86,87,88,89,90,91,92,93,94,95,96,97,98,99]; gB-1 and gB-2 are frequently detected in North America [58,100,101,102,103,104,105,106,107,108,109]; gB-2 is the most extensively sampled genotype in Latin America [110,111,112,113,114,115,116,117,118,119,120,121,122,123]; whereas gB-1, gB-2, and gB-3 are commonly observed across Europe (with the exception of Serbia where gB-4 is the most prevalent genotype) [24,55,57,59,124,125,126,127,128,129,130,131,132,133,134,135,136,137,138,139,140,141,142,143,144,145,146,147,148,149,150,151,152,153]. Although the majority of HCMV-positive individuals who participated in these studies carried gB-1, gB-2, and/or gB-3 genotypes, the association between any particular gB genotype and disease severity is inconsistent among studies [24,55,59,60,61,62,63,64,65,66,67,68,69,70,71,72,73,74,75,76,77,78,83,94,95,96,97,98,99,100,102,103,104,105,106,107,108,109,110,111,112,116,117,118,119,120,121,122,123,124,125,126,127,128,129,130,131,132,133,134,135,136,137,138,139,140,141,142,148,149,150,151,152,153,154]. When a gB genotype has been identified as a disease severity marker, it was often simply the most prevalent genotype in the study population. Thus, it is currently difficult to draw a conclusion as to whether there is a specific gB genotype that leads to a more severe disease progression. Compared to a single gB genotype infection, mixed gB genotype infections are frequently reported in organ/bone marrow transplant recipients and AIDS patients [105,118,125,128,129,152]. Multiple studies have also demonstrated that mixed infections are able to be transmitted in utero [63,68,74,75,76,77,109,122,134,137,138,141]. Although there is no established association between single genotype infections and disease outcomes, patients with mixed gB genotype infections have been reported to have faster disease progression or higher viral loads [83,97,102,144]. This observation highlights the importance of targeting diverse HCMV strains in vaccine design and therapeutics.

gB has been identified as one of the major HCMV vaccine targets due to its essential role in virion fusion with the cell and ability to elicit both neutralizing [155] and non-neutralizing antibody responses [156,157]. Currently, the gB subunit protein from the Towne strain (gB-1) with MF59 adjuvant (gB/MF59) is the most effective vaccine tested to date in several phase I and II clinical trials [158,159,160]. gB-1, gB-2, and gB-4 share a greater genetic similarity compared to gB-3 and gB-5, having been described as comprising two “supergroups” [43]. High-throughput sequencing of viruses from infected individuals who were given either a gB/MF59 or a placebo vaccine showed that gB/MF59 vaccinees were more resistant to infection from HCMV from the gB-1, gB-2, and gB-4 genotype supergroup compared to the placebo recipients, suggesting the protective effect of the vaccine was limited to this supergroup [161]. Additionally, a single variant of gB, 275Y of AD169 strain and 585G of VR1814 strain, was shown to impact viral entry, cell fusion, and genome stability via activation of ATM (ataxia-teleangiectasia mutated) -PIDDosome-caspase-2 signaling axis. Thus, this study suggested that even a single nucleotide polymorphism of gB is critical to the pathogenesis of HCMV infection [162]. Interestingly, with 200 HCMV clinical strains analyzed, gB sequences were found to have a greater diversity among strains compared to those of the pentameric complex. This observation highlights the complexity of gB variation, even though gB genotypes exhibit a ~96% mean conserved identity at the amino acid level. This indicates the dominant action of purifying selection in gB, making it more highly conserved than other glycoproteins including gN and gO [163].

**Figure 1 viruses-13-01106-f001:**
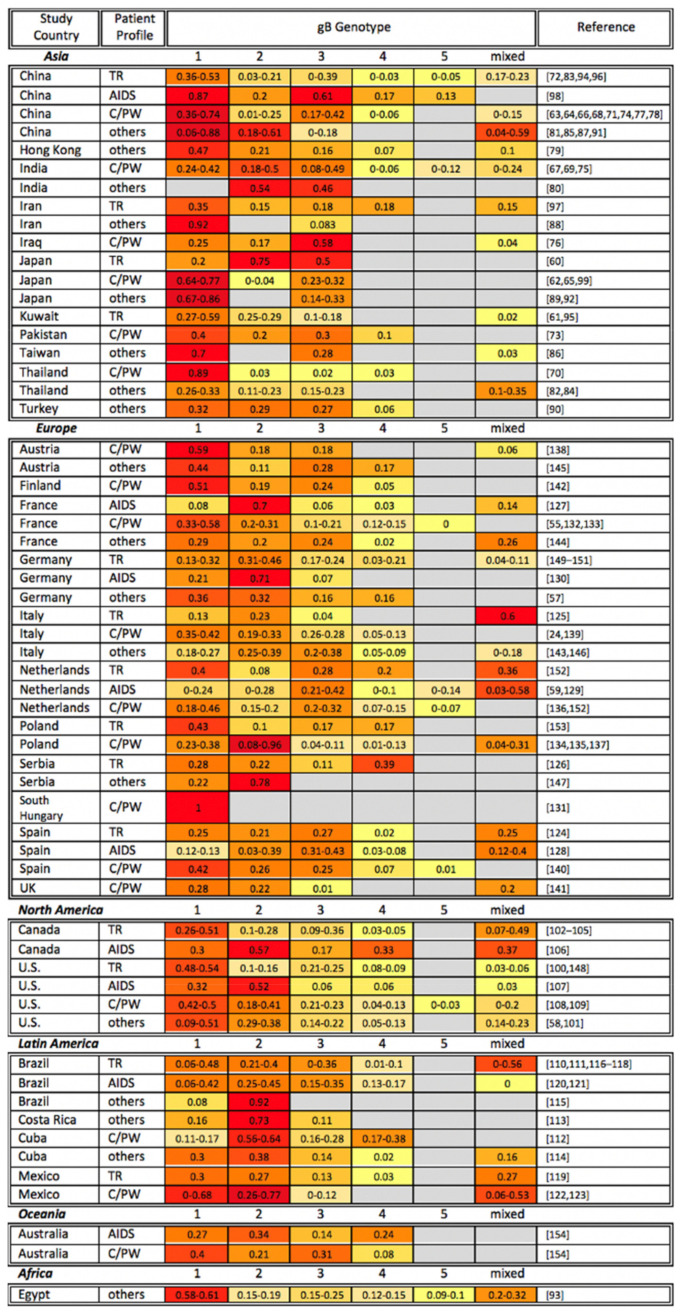
Global gB genotyping across disease states. gB genotyping studies are organized alphabetically by study continent and country, as well as by patient profile (TR: transplant recipient; AIDS: AIDS patient; C/PW: congenital CMV infection or pregnant woman). The percentage of each gB genotype among study participants is shown, with the most common genotypes highlighted in red, genotypes at intermediate frequency in orange, and less common genotypes in yellow. Gray cells indicate that a particular gB genotype was not identified in the reference studies. Note that the sum of all genotypes may be greater or smaller than 1 if multiple gB genotypes were identified in the same sample or if not all samples in a study were genotyped, respectively.

## 5. gN

HCMV gN, encoded by UL73, is a type 1 transmembrane protein that links it to gM by disulfide bonds. Although gN is only composed of ~135 amino acids (NCBI accession number: YP_081521), this protein has aroused interest of late due to its considerable post-translational modifications and high-level of polymorphism. The post-translational modifications of gN include 18 potential phosphorylation sites, 33 possible O-linked glycosylation sites [35], and two N-linked glycosylation sites (reviewed in [164]). gN is highly variable particularly in the N-terminal region, outside virus particles, which is believed to be involved in eliciting immune responses and responding to immunological selective pressure. Based on the differences in the N-terminal region, four gN genotypes have been identified. Two subtypes of gN-3 (gN-3a, gN-3b) and three subtypes of gN-4 (gN-4a, gN-4b, gN-4c) have been further distinguished. The similarity of nucleotides within each genotype is around 80–85%, while the nucleotide similarity within each subtype is 96–100% [35,36].

Considering the geographic distribution of gN genotypes, a global study reported that clinical isolates from China, Australia, and Europe unexpectedly demonstrated a similar frequency of gN-1, gN-3, and gN-4, whereas gN-2 was infrequently detected in Europe and not identified in China and Australia. On the other hand, gN-2 is more commonly detected in North America than other regions, though when detected, gN-1, gN-3, and gN-4 remain the more prevalent [36]. However, the findings of the above global study are not entirely consistent with the local gN genotyping studies as shown in Figure 2. All four gN genotypes have been reported in Asia. However, the gN-4 genotype has often been reported as the most prevalent genotype globally, followed by gN-1 or gN-3, and gN-2. In a single study from Egypt, gN-1 was the most prevalent genotype in breast cancer patients [93].

In terms of the immune responses to gN related to observed genotypic variation, only humoral immunity has been characterized to date. These studies have mainly focused on investigating neutralizing antibody responses against four gN genotypes, and strain-specific neutralizing activity against gN strains has been demonstrated [165,166,167]. It has been shown that anti-gM/gN dimer antibodies possess differential neutralizing activities against AD169, Toledo, and TR strains. Since there have been no gM polymorphisms reported, this study suggested that the neutralizing anti-gM/gN dimer antibody responses against gN might be strain-specific. However, because this study did not isolate gN-specific antibodies and identify neutralizing epitopes lying in the most variable regions of gN, further studies are required to confirm that the neutralization against gN is strain-specific [165]. To investigate the antibody response against gN genotypes more specifically, viruses with four different gN genotypes in the same AD169 virus backbone were constructed. By creating a neutralization assay with these gN-recombinant viruses, strain-specific neutralization was measurable in 30–60% of human sera. Interestingly, human sera collected from subjects in Erlangen (Germany) appeared to neutralize the virus with gN-4 genotype the best, while the cohort from Birmingham (USA) neutralized the virus with gN-2 genotype most potently [166,167]. This observation is consistent with the geographical distribution of HCMV gN genotypes from previous studies [36].

## 6. gO

HCMV gO, a soluble protein encoded by UL74, is an essential element of the gH/gL/gO trimer. Compared to the pentameric complex, the gH/gL/gO trimer has been reported as an integral component of HCMV entry by promoting fusion with all cell types [168]. gO is composed of 457–472 amino acids (NCBI accession number: YP_081522.1), depending on the number of strain-specific deletions. gO is also highly glycosylated, with variation in glycosylation observed among genotypes. The glycosylation sites of gO include 18 potential N-linked glycosylation sites and a single O-linked glycosylation site [50,51]. Of the polymorphisms most commonly observed in the gO sequence, the major variable region lies in the first 98 codons, with some minor variations between codon 270–313 [37]. Based on these differences among gO sequence, four gO genotypes were identified in AIDS patient in 2002 [38], while the fifth gO genotype was verified in 2005 after detection in renal transplant recipients [39]. By tree-based analysis of HCMV sequences in lung transplant recipient samples, three subtypes of gO-1 (gO-1a, gO-1b, gO-1c), and two subtypes of gO-2 (gO-2a, gO-2b) have been reported [40,41].

Previous gO genotyping studies are organized in Figure 3. The neighbor-joining tree analysis of gO genotypes from several studies found disparate genetic similarities between gO genotypes and their subtypes [41,44,169]. In terms of geographic distribution, five gO genotypes have been detected in Asia, Europe, and Australia, with gO-5 being consistently the least frequently detected. Little correlation has been observed between genotype and disease perhaps due to a paucity of gO genotyping studies. Nonetheless, the gO-1 genotype appears to be the most prevalent in most studies [38,41,98,138,170]. Further, previous work has suggested that the gO-1b genotype is linked to gN-3a and gH-1, while the gO-5 genotype was linked to gN-4c and gH-2. However, the implications of this association, and whether it holds any clinical relevance, remains in need of further study [44].

Finally, the functional difference of gO genotypes have been investigated to some extent. Comparing HCMV reconstituted with two highly variable gO genotypes, gO-1 and gO-4, in the same TB40E backbone, the gO-4 genotype displayed an increasing tropism for epithelial cells compared to the gO-1 genotype [171]. Additionally, there was a different inhibitory effect of soluble HCMV trimer- and pentamer-specific entry receptors observed by comparing reconstituted HCMV with gO-1, gO-2, gO-3, and gO-5 genotypes in the same TB40E backbone [172]. It has also been suggested that different gO genotypes have an impact on neutralizing antibody response to gH epitopes [173].

## 7. gH

HCMV gH, encoded by UL75, has been identified as an integral component of the gH/gL/gO trimer and the pentameric complex. In addition to gB, the pentameric complex has remained a key vaccine target given that the complex is required for HCMV to enter epithelial, endothelial, and monocytic cells [30,174,175,176]. Belonging to type I transmembrane protein, gH is 742-amino-acids long (NCBI accession number: YP_081523.1), and compared to other glycoproteins described above, gH is moderately glycosylated and less variable. The sequence of gH has been reported to include six potential N-linked glycosylation sites and four O-linked glycosylation sites [50,51]. Based on the variability of gH sequences, the gH genotypes could be divided into two groups, gH-1 and gH-2. The variable regions between the two groups are mostly identified in the first 37 codons near the N-terminal region and minorly between codons 176–181 and 359–365 [42]. The observed geographic distribution of gH genotypes is shown in Figure 4. Moreover, gH-1 and gH-2 genotypes are distributed fairly equally in organ transplant recipients and HCMV-infected children sampled to date.

gH has been identified as one of the major antigens for eliciting neutralizing antibody responses [177,178]. In a recent study, the neutralizing ability of certain gH-specific monoclonal antibodies were shown to be strain-specific in fibroblasts and epithelial cells. The neutralizing epitopes of these antibodies lie at codons 27–48, the most variable region of gH [47]. Additionally, a previous study has compared the outcomes of renal transplant recipients who elicit matched or mismatched strain-specific gH antibody responses after kidney transplantation. The patients with mismatched anti-gH antibodies appear to have more adverse outcomes compared to those with matched anti-gH antibodies, including a higher possibility of acute tissue rejection, HCMV disease-related manifestations, and a higher level of antigenemia. This suggests that the strain-specific gH antibody may be correlated with disease severity [179]. In terms of cellular immunity, it has also been reported that the T-helper-cell response to gH is strain-specific [48].

## 8. Concluding Remarks

We have reviewed the commonly observed polymorphic sites and regions of HCMV glycoproteins (gB, gN, gO, and gH), which play an essential role in the formation of glycoprotein complexes on the surface of the HCMV virion. Additionally, we have organized data from previous global genotyping studies of HCMV glycoproteins. The dominant gB and gH genotypes across disease states in each continent are summarized in Table 3. The percentage of each genotype in the same country varies by study, and further data will be required to determine whether significant geographic statistical associations exist.

We have also reviewed the biological relevance of polymorphisms within viral glycoproteins by presenting evidence relating them with strain-specific immune responses. The biological relevance of specific genotypic variation has been suggested by several studies. For example, gH neutralizing epitopes were shown to lie in the most variable regions of gH [47], suggesting that the polymorphisms of glycoproteins might be an effective strategy for virus immune evasion. Additionally, instead of directly mediating neutralizing antibody responses, the polymorphisms of glycoproteins could indirectly moderate neutralizing antibody responses by blocking neutralizing epitopes of other glycoproteins. For example, it has been reported that polymorphisms within gO protected neutralizing epitopes of gH [173]. In terms of strain-specific T-cell immunity, there are currently no studies at the individual epitope level relating to glycoprotein variation, though one study has reported that T-helper cell response to gH is strain-specific [48]. Further studies are required to fully understand the functional relevance of such glycoprotein variability.

Although considerable data have been accumulated in recent years, many important questions still remain regarding how common polymorphisms in glycoproteins should best be incorporated into CMV vaccine development. Specifically, the nature and degree of associations between genotypic variation in the glycoproteins and functional difference in HCMV, as well as the elicited strain-specific immunity by genotype, should be further studied in relation to protection against CMV acquisition and reinfection. Although the situation is further complicated by the prevalence of mixed-strain infections, the relationship between these mixed infection and disease severity and progression is of growing interest. These emerging results suggest the importance of multivalent vaccine designs, as well as a consideration of glycoprotein variation in future HCMV vaccine design and therapy.

## Figures and Tables

**Figure 2 viruses-13-01106-f002:**
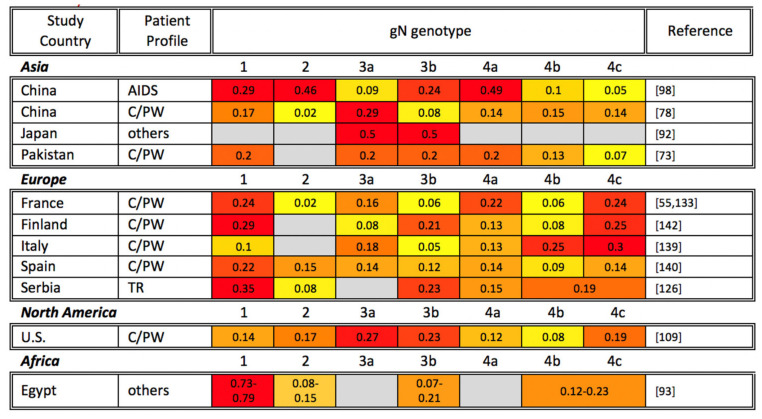
Global gN genotyping across disease states. gN genotyping studies are organized alphabetically by study continent and country, as well as by patient profile (TR: transplant recipient; AIDS: AIDS patient; C/PW: congenital CMV infection or pregnant woman). The percentage of each gN genotype among study participants is shown, with the most common genotypes highlighted in red, genotypes at intermediate frequency in orange, and less common genotypes in yellow. Gray cells indicate that a particular gN genotype was not identified in the reference studies. Note that the sum of all genotypes may be greater or smaller than 1 if multiple gN genotypes were identified in the same sample or if not all samples in a study were genotyped, respectively.

**Figure 3 viruses-13-01106-f003:**
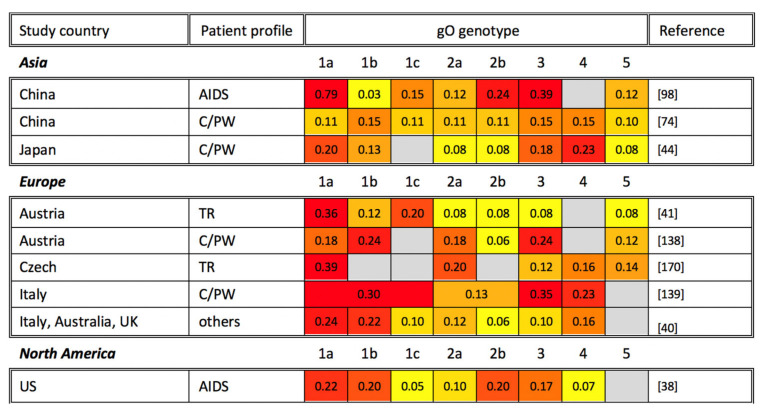
Global gO genotyping across disease states. gO genotyping studies are organized alphabetically by study continent and country, as well as by patient profile (TR: transplant recipient; AIDS: AIDS patient; C/PW: congenital CMV infection or pregnant woman). The percentage of each gO genotype among study participants is shown, with the most common genotypes highlighted in red, genotypes at intermediate frequency in orange, and less common genotypes in yellow. Gray cells indicate that a particular gO genotype was not identified in the reference studies. Note that the sum of all genotypes may be greater or smaller than 1 if multiple gO genotypes were identified in the same sample or if not all samples in a study were genotyped, respectively.

**Figure 4 viruses-13-01106-f004:**
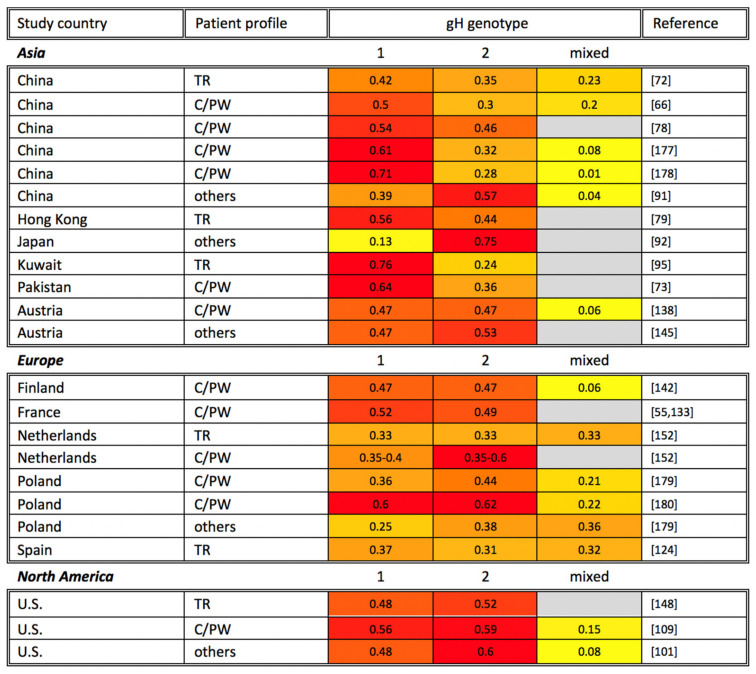
Global gH genotyping across disease states. gH genotyping studies are organized alphabetically by study continent and country, as well as by patient profile (TR: transplant recipient; AIDS: AIDS patient; C/PW: congenital CMV infection or pregnant woman). The percentage of each gH genotype among study participants is shown, with the most common genotypes highlighted in red, genotypes at intermediate frequency in orange, and less common genotypes in yellow. Gray cells indicate that a particular gH genotype was not identified in the reference studies. Note that the sum of all genotypes may be greater or smaller than 1 if multiple gH genotypes were identified in the same sample or if not all samples in a study were genotyped, respectively.

**Table 1 viruses-13-01106-t001:** Commonly observed glycoprotein polymorphic regions.

Glycoprotein	Length (Amino Acids)	Genotypes (Subtypes)	Most Variable Region	References
gB (UL55)	907	gB-1, gB-2, gB-3, gB-4, gB-5	codons 26–70, gp55 cleavage site (codon 460)	[32,33,34]
gN (UL73)	135	gN-1, gN-2, gN-3 (gN-3a, gN-3b), gN-4 (gN-4a, gN-4b, gN-4c)	N-terminal region (codons 1–87)	[35,36]
gO (UL74)	457–472	gO-1 (gO-1a, gO-1b, gO-1c), gO-2 (gO-2a, gO-2b), gO-3, gO-4, gO-5	N-terminal region (codons 1–98), codons 270–313	[37,38,39,40,41]
gH (UL75)	743	gH-1, gH-2	N-terminal region (codons 1–37)	[42]

**Table 2 viruses-13-01106-t002:** Genotyping of common laboratory-adapted HCMV strains.

Virus Strain	gB (UL55) ^1^	gN (UL73) ^2^	gO (UL74) ^3^	gH (UL75) ^4^	References
Towne	gB-1	gN-4b	gO-4	gH-2	[33,36,40,42]
TB40E	gB-1	gN-4c	gO-1c	gH-1	[36,40]
AD169	gB-2	gN-1	gO-1a	gH-1	[33,36,40,42]
Toledo	gB-3	gN-4c	gO-1c	gH-1	[36,38,43]
VR1814	gB-3	gN-4c	gO-1c	gH-1	N/A
Merlin	gB-1	gN-4c	gO-5	gH-2	[44]

Genotypes were determined by multiple sequence alignment to genotype-specific reference sequences (Appendix A) using T-coffee version 13.45.0.4846264 [31]. ^1^ The GenBank accession numbers of published gB sequences are: TB40E strain (ABV71586.1), VR1814 strain (ACZ79977.1), and Merlin strain (AAR31620.1). ^2^ The GenBank accession numbers of published gN sequences are: VR1814 strain (ACZ79984.1) and Merlin strain (F5HHQ0.1). ^3^ The GenBank accession number of the published gO sequence is: VR1814 strain (ACZ79985.1). ^4^ The GenBank accession numbers of published gH sequences are: TB40E strain (ABV71597.1), VR1814 strain (ACZ79986.1), and Merlin strain (YP_081523.1).

**Table 3 viruses-13-01106-t003:** Dominant gB and gH genotypes across disease states in each continent.

**Transplant Recipient**				
***Continent***	***Dominant gB*** ***Genotype ^1^***	***References***	***Dominant gH*** ***Genotype ^1^***	***References***
Asia	gB-1, gB-2	[60,61,72,83,94,95,96,97]	gH-1	[72,79,95]
Europe	gB-1, gB-2	[124,125,126,149,150,151,152,153]	gH-1, gH-2	[124,152]
North America	gB-1	[100,102,103,104,105,148]	gH-1, gH-2	[148]
Latin America	gB-1, gB-2	[110,111,116,117,118,119]		
Oceania		N/A		N/A
		**AIDS patient**		
***Continent***	***Dominant gB*** ***Genotype ^1^***	***References***		
Asia	gB-1	[98]		
Europe	gB-2	[59,127,128,129,130]		
North America	gB-2	[106,107]		
Latin America	gB-1, gB-2, gB-3	[120,121]		
Oceania	gB-1, gB-2, gB-4	[154]		
**Congenital CMV infection or pregnant woman**				
***Continent***	***Dominant gB*** ***Genotype ^1^***	***References***	***Dominant gH*** ***Genotype ^1^***	***References***
Asia	gB-1	[62,63,64,65,66,67,68,69,70,71,73,74,75,76,77,78,99]	gH-1	[66,73,78,138,180,181]
Europe	gB-1	[24,55,131,132,133,134,135,136,137,138,139,140,141,142,152]	gH-1, gH-2	[55,133,142,152,182,183]
North America	gB-1, gB-2	[108,109]	gH-1, gH-2	[109]
Latin America	gB-2	[112,122,123]		
Oceania	gB-1	[154]		

^1^ The dominant gB or gH genotype was determined by identifying the most common gB or gH genotype across all referenced studies.

## Data Availability

Not applicable.

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
