# Peer review of "Common Polymorphisms in the Glycoproteins of Human Cytomegalovirus and Associated Strain-Specific Immunity"

_viruses, 2021, doi:10.3390/v13061106_

Round 1

Reviewer 1 Report

Understanding the glycoprotein polymorphism and its impact on human immunity against HCMV infection is  important to develop an effective HCMV vaccine. The authors provide a thorough review of polymorphisms of some of the HCMV glycoproteins (i.e. gB, gN, gO, and gH), the geographic distribution of the different genotype/subtypes, as well as the reported impact on neutralizing activity against different strains/isolates of HCMV as it relates to the corresponding glycoproteins polymorphisms. The manuscript represents a good and clear summary of the literature. Below are a couple of minor suggestions:

  1. Provide an additional/supplemental table showing the defining mutations for each genotype/subtype of glycoproteins
  2. Incorporate VR1814 and Merlin into Table 2, as both viruses are also commonly used in the field.

Author Response

Here is my point-to-point reply to reviewer 1’s specific comments:

  1. Provide an additional/supplemental table showing the defining mutations for each genotype/subtype of glycoproteins.
  • We have included 4 supplementary figures to show the defining mutations for each glycoprotein.
  1. Incorporate VR1814 and Merlin into Table 2, as both viruses are also commonly used in the field.
  • We have added the glycoprotein genotyping info of VR1814 and Merlin strains into Table 2.

Reviewer 2 Report

General comments:

This review summarizes the different CMV genotypes defined by polymorphisms in immunologically important viral envelope glycoproteins and their prevalences in different countries and patient populations as reported by numerous independent studies. Overall, this is a clearly written, well organized, and concise review. However, it somewhat neglects the “big picture” biological relevance of polymorphisms within this set of viral glycoproteins. The comments below suggest some areas that the authors might consider expanding on as topics for Discussion.   

Thus far there appears to be limited evidence for disease associations with specific CMV genotypes or to indicate that certain genotypes predominate in certain geographical regions but not in others. While figures 1-4 show that throughout the world some genotypes are common while others are less so, there is a great deal of study-to-study variation and the extent to which genotype prevalences vary between different studies from the same region or country seems to suggest that there is no statistical significance to assertions that certain genotypes are more prevalent in certain geographical regions compared to others. If this is not the case, then the statistical analyses need to be presented and discussed. If it is the case, then it should be stated more clearly.

Presumably they arose under selective pressure to evade host immunity and now perhaps provide some partial advantage to a strain with one set of polymorphisms in overcoming immunity in a host previously infected by a different strain. The gH epitope described on lines 282-284 seems to be an example of this – a neutralizing epitope in gH has two “alleles” and there are monoclonal antibodies that effectively neutralize viruses encoding one allele but have no activity against viruses that encode the other allele. It would be very helpful to review whether there are similar examples of polymorphisms in other glycoproteins that dictate strain specificity not just for antibody binding, but more importantly, neutralizing activity. I am not aware of data on strain polymorphisms dictating T-cell functions at the individual epitope level, but presumably this could also occur.

In addition to polymorphisms that directly impact neutralizing epitopes, there are now two examples of polymorphic proteins acting to protect neutralizing epitopes that lie within other proteins in a complex. A role for gO in this regard is briefly mentioned in lines 255-256, but a similar phenomenon has also been observed for gN. This would seem to provide particularly powerful evidence for biological relevance as some polymorphic proteins appear to have evolved specifically to protect other epitopes from neutralizing antibodies.

Visually it might be helpful to show cumulative genotype distributions for all the studies from Asia, Europe, etc., and perhaps compare to the global average from all the studies. Similarly, it might be useful to group patient types (transplant, AIDS, congenital, other) together regardless of geographical location.

It would also be very helpful to illustrate the nature of the polymorphisms in question using amino acid alignments that include sequences representing each of the subgenotypes. Such alignments can nicely demonstrate the clustering of polymorphic regions within each protein and the extent of sequence diversity as well as how the different genotypes cluster together in groups that are highly similar within a group yet quite distinct from others (i.e., there is not much “gradient” in between).

Specific comments:

  1. Line 64. The grammar police would argue that “encodes” means ”codes for”, so “encodes for” is redundant. While I am not necessarily opposed to the occasional redundancy, my preference is for “encodes”.
  2. Line 66. As worded, “…responsible for viral attachment…” implies that all 65 glycoproteins are responsible for attachment or entry. While the functions of most of these glycoproteins remain uncertain, it is likely that many are not involved in attachment or entry.
  3. Tables 1 and 2 should have a list of references that defined the subtypes that are listed.
  4. Table 2. The designations gC-I, gC-II, and gC-III are not explained. gC-II includes gM and gC-III presumably includes gL. Strain Merlin should probably be included here. Why are there no gH genotypes for TB40E and Toledo? While perhaps the source reference did not include them, it should be possible for the authors to determine the appropriate gH genotypes based on their gH sequences.
  5. Line 105. It is unclear what “minor coding regions” means.
  6. Line 116. “by enlarge” appears to be a typo of “by and large”.
  7. Line 141. “fusion to the cell” should perhaps be “fusion with the cell”.
  8. Line 155. “gp1” is not mentioned or described elsewhere. How is gp1 relevant?
  9. Fig. 2. US data are mislabeled as “Europe”
  10. Line 206. While perhaps there have been no publications that discuss gM polymorphisms, gM has ~10 single amino acid positions that are polymorphic and one block from 286 to 295 that appear to comprise at least two cluster groups.
  11. Line 220. The cited reference suggests that the trimer is important for entry into all cell types, including epithelial cells and presumably others.
  12. Line 231. Unclear what is meant by “…disparate genetic similarities…”  
  13. Line 282. As worded, this seems to imply that all antibodies to gH are strain specific. Suggest “…certain gH-specific antibodies…”
